# Anticipation of ventricular tachyarrhythmias by a novel mathematical method: Further insights towards an early warning system in implantable cardioverter defibrillators

**Gabriel S. Zamudio**[1]\*, **Manlio F. Márquez**[2], **Marco V. José**[1]\*

**1** Theoretical Biology Group, Instituto de Investigaciones Biomédicas, Universidad Nacional Autónoma de México, Ciudad de México, México, **2** Electrophysiology Department, Instituto Nacional de Cardiología Ignacio Chávez, Mexico City, Mexico

\* gazaso92@gmail.com (GSZ); marcojose@biomedicas.unam.mx (MVJ)

**Data Availability Statement:** The data underlying this study are third party. The data was downloaded from the Spontaneous Ventricular

## Abstract

Implantable cardioverter defibrillators (ICD) are the most effective therapy to terminate malignant ventricular arrhythmias (VA) and therefore to prevent sudden cardiac death. Until today, there is no way to predict the onset of such VA. Our aim was to develop a mathematical model that could predict VA in a timely fashion. We analyzed the time series of R-R intervals from 3 groups. Two groups from the Spontaneous Ventricular Tachyarrhythmia Database (v 1.0) were analyzed from a set of 81 pairs of R-R interval time series records from patients, each pair containing one record before the VT episode (Dataset 1A) and one control record which was obtained during the follow up visit (Dataset 1B). A third data set was composed of the R-R interval time series of 54 subjects without a significant arrhythmia heart disease (Dataset 2). We developed a new method to transform a time series into a network for its analysis, the $\varepsilon-regular\ graphs$. This novel approach transforms a time series into a network which is sensitive to the quantitative properties of the time series, it has a single parameter ($\varepsilon$) to be adjusted, and it can trace long-range correlations. This procedure allows to use graph theory to extract the dynamics of any time series. The average of the difference between the VT and the control record graph degree of each patient, at each time window, reached a global minimum value of −2.12 followed by a drastic increase of the average graph until reaching a local maximum of 5.59. The global minimum and the following local maxima occur at the windows 276 and 393, respectively. This change in the connectivity of the graphs distinguishes two distinct dynamics occurring during the VA, while the states in between the 276 and 393, determine a transitional state. We propose this change in the dynamic of the R-R intervals as a measurable and detectable "early warning" of the VT event, occurring an average of 514.625 seconds (8:30 minutes) before the onset of the VT episode. It is feasible to detect retrospectively early warnings of the VA episode using their corresponding $\varepsilon-regular\ graphs$, with an average of 8:30 minutes before the ICD terminates the VA event.

Tachyarrhythmia Database Version 1.0 from Medtronic Inc. (available at doi.org/10.13026/C25K5D). A set of 81 pairs of R-R interval time series records from different patients was obtained (Group 1A), each pair contains one record before the VT episode and one control record (CR) which was obtained during the follow up visit (Group 1B). A third data set composed of the R-R interval time series of 54 subjects without a significant arrhythmia heart disease was obtained (available at doi.org/10.13026/C2NK5R). The authors did not have special access privileges.

**Funding:** Gabriel S. Zamudio is a doctoral student from Programa de Doctorado en Ciencias Biomédicas, Universidad Nacional Autónoma de México (UNAM) and a fellowship recipient from Consejo Nacional de Ciencia y Tecnología (CONACYT) (number: 737920). Marco V. José was financially supported by PAPIIT-IN201019, UNAM, México.

**Competing interests:** The authors have declared that no competing interests exist.

## Introduction

Implantable cardioverter defibrillators (ICD) are the cornerstone of sudden cardiac death prevention through termination of ventricular tachycardia/ventricular fibrillation. Although ICD shocks usually occur when the subject is unconscious, it could be very useful to patients and close relatives to have the possibility to know in advance, either seconds or minutes, when those malignant arrhythmias could occur in order to take appropriate preventive measures. We hypothesized that a novel mathematical analysis, $\varepsilon-regular\ graphs$, could perform such task.

Network theory possesses the capacity to abstractly represent interactions of any kind of entities. Currently, complex networks have arisen as a common way to tackle intricate dynamics [1]. A broad range of applications in different biological and medical areas abound. In the area of biology, they have been used to analyze a population's structure [2, 3], and pandemics [4, 5]. The use of protein-protein interaction networks coupled with information theory have led to discover potential therapeutic biomarkers on cancer research [6]. Integrative approaches for anticipating critical transitions have been proposed [7] in several phenomena, although the area of cardiology has not yet been explored. The terms "early warnings" and "tipping points" are still not part of the cardiologist community. Several methods have been developed to transform time series into networks for its analysis. Such methods include the visibility graphs method [8], and a plethora of its modifications [9, 10], which consider the topological properties of the time series, the recurrence analysis of time series [11, 12], and the analysis based on the phase space [13]. In this work, we usher in a new method to transform a time series into a network for its analysis, the $\varepsilon-regular\ graphs$. This novel approach transforms a time series into a network which is sensitive to the quantitative properties of the time series, it has a single parameter ($\varepsilon$) to be adjusted, and it can capture long-range correlations. This procedure permits using graph theory to extract the dynamics of any time series. As a direct application of $\varepsilon-regular\ graphs$, data from patients diagnosed with imminent ventricular tachyarrhythmias (VT) was analyzed. The heart activity is driven by the action of the opposing forces of the sympathetic and the parasympathetic nervous systems [14, 15]. The failure in heart function is the result of malfunctions in the myocardium, heart valves, pericardium, or the endocardium [16].

### Limitations

The method proposed in this work requires further testing with patients whose clinical history is well documented and controlled, coupled with respiratory data. A significant clinical limitation of this work is the fact that this approach is restricted to patients with normal sinus rhythm and is unlikely to work in patients with atrial fibrillation or those with a pacemaker. In the former group because of the large variability of R-R intervals, and in the later because of fixed pacing rhythms.

### Methods

#### A mathematical method to transform time series to networks

The method consists in assigning to each point in a time series a vertex in the network. Then, for a fixed value of the parameter $\varepsilon = \varepsilon_0$, any two points of the time series $p_1, p_2$ will be joined in the network, if and only if $|p_1-p_2| \leq \varepsilon_0$; this means that two point of the time series will be adjacent in the $\varepsilon-regular\ graph$ if the values of the points have a maximum difference of $\varepsilon_0$. For illustrative purposes, we show a diagram of the algorithm in Fig 1. In Fig 1A, we show a time series of ten points; the values of the points are $p_1, p_7$, and $p_{10} = 0.2$; $p_2$ and $p_4$ are equal to 0.29; $p_5 = p_8 = 0.38$; $p_3 = p_6 = p_9 = 0.7$. The $\varepsilon-regular\ graph$ is constructed with a parameter

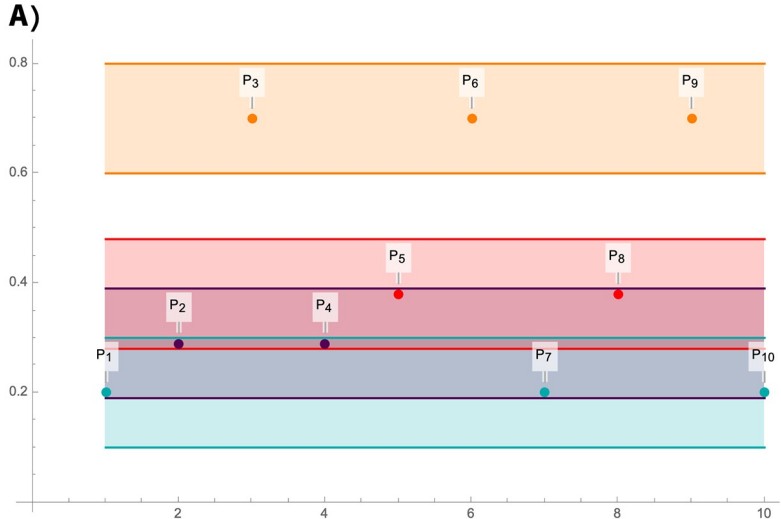

**B)**

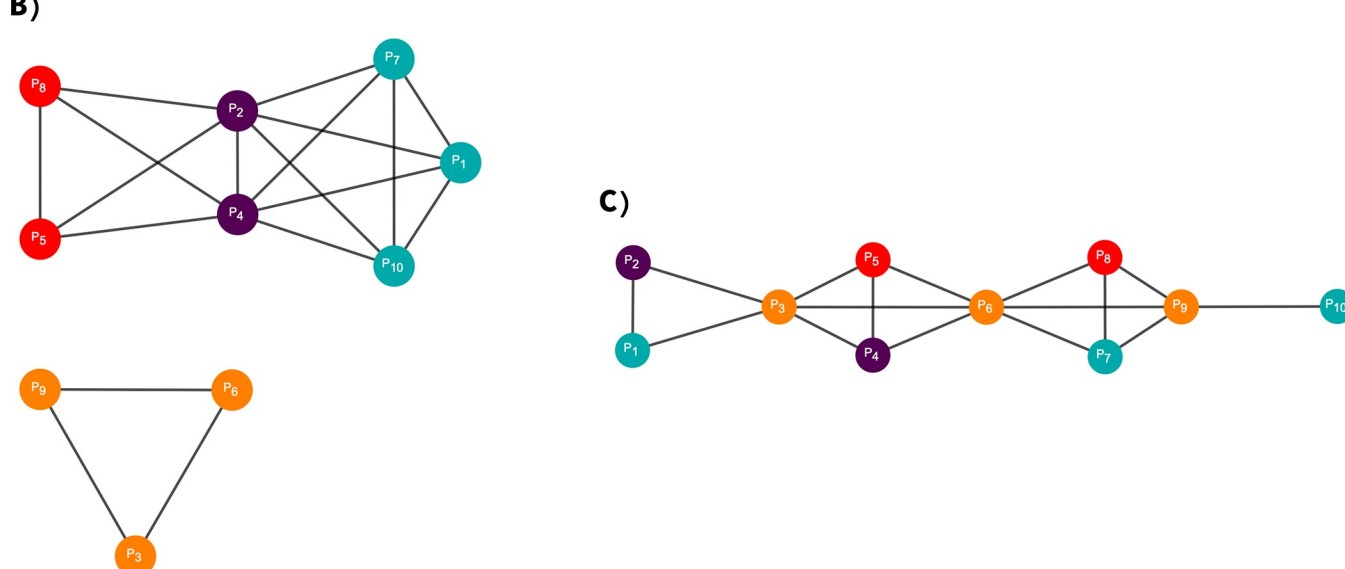

**Fig 1. Diagram of the $\varepsilon$−graph algorithm.** In (A), a time series of ten points from which an $\varepsilon$−regular graph is constructed with a parameter value of $\varepsilon_0 = 0.1$. In (B), intervals of width $\varepsilon_0 = 0.1$ are drawn around each point of the time series. For a given point $p$ of the time series, all the point lying inside the interval of width $\varepsilon_0$ will be adjacent to $p$ in the corresponding $\varepsilon$−regular graph. Note that points with a higher value of 0.7, belong to a different component than the rest of time series reflecting its outlier nature. Also note that the periodic values, $p_3, p_6$ and $p_9$ form a subgraph. Points with the same value that are not periodic will form a complete subgraph, such as the points $p_1, p_7$ and $p_{10}$. In (C), the time series is transformed to a graph using the visibility-graph algorithm.

value of $\varepsilon_0 = 0.1$. In Fig 1B, intervals of width $\varepsilon_0 = 0.1$ are drawn around each point of the time series. For a given point $p$ of the time series, all the point lying inside the interval of width $\varepsilon_0$ will be adjacent to $p$ in the corresponding $\varepsilon$−regular graph.

Other algorithms to convert time series into graphs have been developed but with qualitative instead of quantitative rules for determining the adjacencies in the corresponding graphs (visibility plots). The algorithm for the visibility graphs determines the adjacencies by analyzing the lines joining the points of the time series [8] as observed in Fig 1C. The visibility graph algorithm confers to its graph properties that strongly differ to our $\varepsilon$−regular graph derived from the same time series because the former does not capture the quantitative properties of a

time series as the $\varepsilon-regular\ graph$ do. When considering the outliers of a time series, in a visibility graph, the extremely high or low values of a time series would be "visible" from almost all the rest of the points of the time series, and thus, according to the visibility graph algorithm would be highly connected and act as a hub. This would remain true even if the outlier value were not so extremely contrasting. In contrast, an outlier in a $\varepsilon-regular\ graph$ would have different behaviors according to its value. If the outlier value is significantly different, it will be assigned to a different component in the $\varepsilon-regular\ graph$: If the suspected outlier's point value is not so different to the rest of the time series, it will remain in the same component. This behavior is reflected in the time series of Fig 1, where points with a higher value of 0.7, belong to a different component than the rest of time series reflecting its outlier nature. If instead of the value 0.7, the values were set to 0.48, they will still be higher than the rest of the time series but will remain in the same graph component of the rest of the time series points since the $\varepsilon_0$ value is set at $\varepsilon_0 = 0.1$, and the points with the second-highest value are equal to 0.38.

From the definition of adjacencies in an $\varepsilon-regular\ graph$, it is directly derived that on a given set of points, in which the points of the set have a value difference up to $\varepsilon_0$ among them, they will be joined and thus they will form a complete subgraph. This result is useful when considering time series with regular or periodic values. The periodic values will form a complete subgraph, as the points $p_3, p_6$, and $p_9$ in Fig 1B. Also, points with the same value that are not periodic will form a complete subgraph, such as the points $p_1, p_7$, and $p_{10}$ (Fig 1) that form the complete graph $K_3$. A similar property is not inherited in visibility graphs. In visibility graphs, periodic or regular points from the time series might in some cases not be adjacent, as some points might block the visibility condition for them to be adjacent. What can be rescued from visibility graphs is the short-term correlations of the time series.

As proof of concept, two time series were simulated from theoretical frameworks and their corresponding $\varepsilon-regular\ graphs$ were derived. Time series of $10^4$ points were simulated, the first time series was obtained from a standard normal probability distribution, and the second from a standard Brownian motion valued at integer times. The value of the parameter $\varepsilon_0$ was set as the standard deviation, and half of the standard deviation of both the normal distribution and the Brownian motion, which correspond to 1 and 1/2 respectively, in both cases. The degree centrality measure was calculated for the resulting graphs (Fig 2). The values of the centrality measure were standardized to the unit interval. The degree distributions for the graphs constructed form the Brownian motion approximates a Gaussian curve despite its multiple modes; and the distribution from points sampled from a normal distribution approximates a lognormal distribution. The distributions of the degree measurements maintain, up to some extent, the statistical properties from the time series they were derived from. The difference between any two values of the Brownian motion follows a normal distribution, and this property is shown by the $\varepsilon-regular\ graphs$ in the degree centrality measure when setting the $\varepsilon-$parameter equal to values related to the standard deviation.

The computer program is found in Supporting Information I.

## Application to imminent VT

As a direct application of the $\varepsilon-regular\ graphs$ algorithm, a set of time series related to VT was analyzed. The data was downloaded from the Spontaneous Ventricular Tachyarrhythmia Database Version 1.0 from Medtronic Inc. (available at http://physionet.org/physiobank/database/mvtdb/) [17]. A set of 81 pairs of R-R interval time series records from different patients was obtained (Group 1A), each pair contains one record before the VT episode and one control record (CR) which was obtained during the follow up visit (Group 1B). A third data set composed of the R-R interval time series of 54 subjects without a significant

# Degree centrality of the applied ε- regular graph algorithm

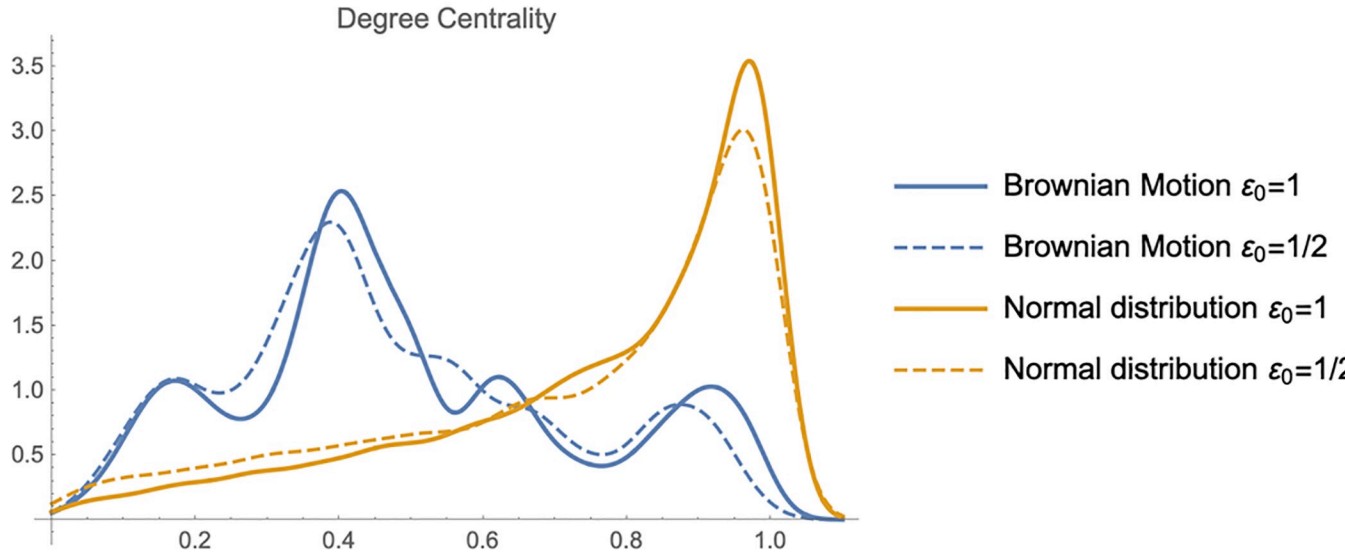

**Fig 2. Degree centrality of the applied $\varepsilon-$graph algorithm.** Two time series were simulated from theoretical frameworks and their corresponding ε-regular graphs were derived. Time series of 10,000 points were simulated, the first time series was obtained from a standard normal probability distribution (solid and dashed yellow curves), and the second from a standard Brownian motion (solid and dashed blue curves) valued at integer times. The value of the parameter $\varepsilon_0$ was set as the standard deviation, and half of the standard deviation of both the normal distribution and the Brownian motion, which correspond to 1 and 1/2, respectively, in both cases. The degree distributions for the graphs constructed form the Brownian motion approximates a Gaussian curve despite its multiple modes; and the distribution from points sampled from a normal distribution approximates a lognormal distribution.

arrhythmia heart disease was obtained (available at: https://physionet.org/physiobank/database/nsrdb/) [16]. This third dataset of time series will be hereafter considered as healthy subjects (HS) and denoted as Group 2.

The time series for groups 1A and 1B were cropped to the same length starting from the end to make them directly comparable (985 points), the start of the VT episode is at the last point of the time series. The Group 2 (HS) series were also cropped by subsampling a random set of sequential points of each subject that match the length of the VT and CR time series. The VT, CR, and HS time series of each patient were subdivided in time series of 60 points with a sliding window method with an offset of one point. This would allow analyzing the change of the series in time to detect an early warning of the VT episode. The time series on each window were transformed into graphs with the $\varepsilon-regular\ graph$ algorithm setting the parameter value at $\varepsilon_0 = 0.04$.

The average degree of the graphs from each window of each subject was calculated and averaged among the subjects, for the three different time series. This process results in a time series of the average degree of the graphs representing the three different states of the subjects (Fig 3). The datasets VT and CR arise from the same subjects, so a direct comparison of subjects prior to a VT episode and in a normal stage is possible. The average of the difference between the VT and the CR graph degree of each patient, at each time window (Fig 3), reaches a global minimum value of −2.12, followed by a drastic increase of the average graph until reaching a local maximum of 5.59. The global minimum and the following local maxima occur at the windows 276 and 393, respectively. This change in the connectivity of the graphs distinguishes two distinct dynamics occurring during the ventricular tachyarrhythmia, while the states in between the 276 and 393, determine a transitional state. We propose this change in

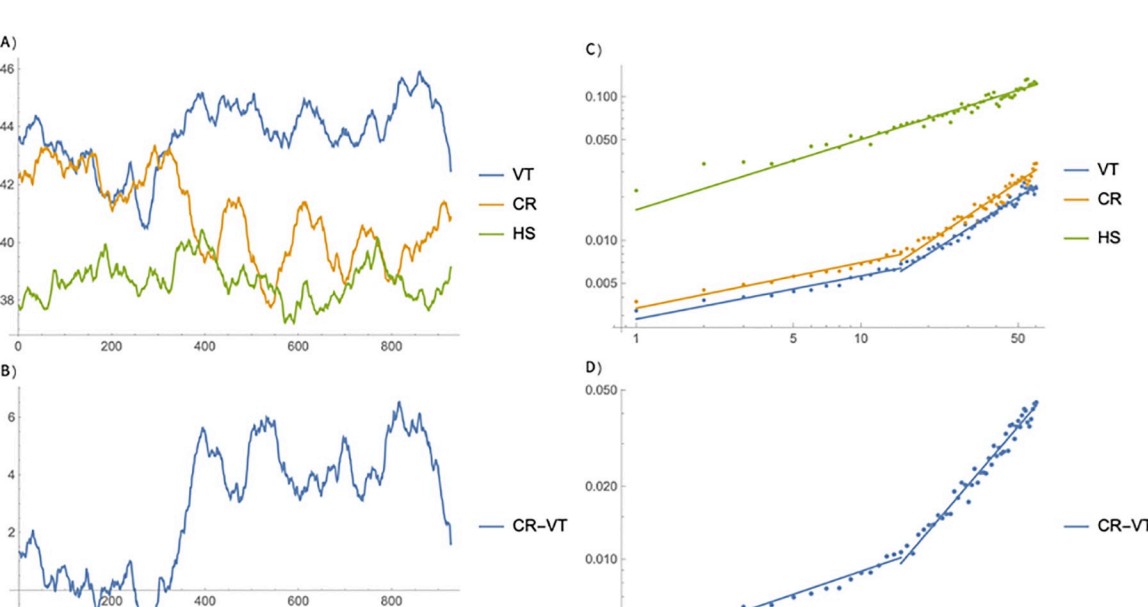

**Fig 3. Early warning of a VT event.** Comparison of the $\varepsilon$−graph algorithm and detrended fluctuation analysis. In (A), a set of 81 pairs of RR interval time series records from different patients was obtained (Group 1A), each pair contains one record before the VT episode and one control record (CR yellow solid curve) which was obtained during the follow up visit (Group 1B). A third data set composed of the RR interval time series of 54 healthy subjects (HS green solid curve) without a significant arrhythmia heart disease was obtained, denoted also as Group 2. In (B), the result of applying the detrended fluctuation analysis (DFA) to the RR series displayed in (A), are shown. The start of the VT episode is at the last point of the time series. The time series on each window were transformed into $\varepsilon$−graphs setting the parameter value at $\varepsilon_0 = 0.04$. The average degree of the graphs from each window of each subject was calculated and averaged among the subjects, for the three different time series. This process results in a time series of the average degree of the graphs representing the three different states of the subjects In (C), the average of the difference between the VT and the CR graph degree of each patient, at each time window reaches a global minimum value of -2.12, followed by a drastic increase of the average graph until reaching a local maximum of 5.59. The global minimum and the following local maxima occur at the windows 276 and 393, respectively. This change in the connectivity of the graphs distinguishes two distinct dynamics occurring during the ventricular tachyarrhythmia, while the states in between the 276 and 393, determine a transitional state. In (D), the corresponding DFA of the curve obtained in (C) is shown.

the dynamic of the R-R intervals as a measurable and detectable early warning of the VT event, occurring an average of 514.625 seconds (8:30 minutes) before the start of the VT episode. The count of 514.625 seconds corresponds to the sum of the average of the time lapses of the last R-R intervals starting from the point 276 to when the VT episode begins at point 985.

The optimization of the parameter $\varepsilon$ is based on the statistical parameters of the R-R time series. The average, minimum and maximum distance between two points of the R-R time series are: 0.043, $1.11 \times 10^{-16}$, 0.12, respectively. Thus, the $\varepsilon$ value of 0.04 approximates the mean of the differences. In Supporting Information II, we show the degree time series when the values 0.01, 0.02, 0.03, 0.04, and 0.06 are assigned to the parameter $\varepsilon$ (S1 Fig). Note that the overall pattern is preserved. In particular, the abrupt change of the dynamics is captured by the different values of the parameter $\varepsilon$.

## Comparison with detrending fluctuation analysis

A widely used procedure used in the analysis of data originated from diverse heart records is the application of the Detrended fluctuation analysis (DFA) method. DFA is a mathematical

linear method to analyze time series by removing the linear trend of time series divided into smaller windows. This method is of special use to address nonstationary time series [18]. Results from the DFA method are commonly graphed in a log-log scale and the scaling exponent of the time series is estimated from a least-square fit of a linear model. The scaling exponent measures the correlation in the noise and approximates the Hurst exponent of the time series.

The DFA method was applied to the mean time series of the 3 groups (Fig 3A), and the difference between the mean VT and CR subjects (Fig 3B). From the DFA of the different states of the subjects it can be observed that the time series from the HS subjects possess the same scaling properties at short- and long-time lengths, which is deduced from the fact that the DFA approximates a linear model. On the other hand, the DFA analysis from the VT and CR subjects show that their time series possess two different scales of autocorrelation, which is related to the two different linear models fitting the DFA of VT and CR. The VT and CR time series behave similarly in the sense that both exhibit different scaling properties for short correlations (windows with 15 or less points) and a different scaling factor for long correlations (15 or more points). Since the slope of the liner models fitted to VT, 0.24 and 0.95 for short and long correlations, respectively, and CR, 0.25 for short and 1.01 for long correlations (Fig 3C), are practically the same, then, their corresponding Hurst exponents will be the same, which results in that the VT and CR behave similarly regardless if short or long correlations are assessed. This is validated by the fact that the slopes of the two linear models fitted to the DFA of VT-CR, 0.31 and 1.08 for short and long correlations, approximates the ones obtained when analyzing VT and CR separately (Fig 3D). The DFA method is capable to discern the different scaling properties occurring on the Groups 1A (VT) and 1B (CR) patients as compared to the Group 2 (HS) subjects. However, the DFA results are not varying in time, and hence this method is not capable of discerning an early warning for VT.

## Discussion

In this work we propose a novel parametric method to analyze time series by transforming them into networks. By using this method, it is possible to apply the graph and network theory in the analysis of time series. Herein, a direct application of the $\varepsilon-regular\ graph$ method is herein shown by using time series data derived from patients with ventricular heart tachyarrhythmia disease. The application of the $\varepsilon-regular\ graph$ method, using a sliding window framework, detected a potential early warning of the disease that it is not detectable using the current linear methods available for the analysis of time series. The $\varepsilon-regular\ graphs$ differ from the visibility graph method as the former is a parametric quantitative method and the latter is a qualitative approach. The adjustable parameter $\varepsilon$ in $\varepsilon-regular\ graphs$, determines the sensitivity of the transformation of time series to networks. By varying the parameter, it is possible to obtain a range of graphs going from graphs in which a vertex is only connected to other ones having the same value, up to completely connected graphs. An inverse transformation, form a network to a time series, would be possible if there exists a compendium of graphs derived from the same time series using different $\varepsilon$ values. Then, if needed, the original time series can be inferred using the different adjacencies from the $\varepsilon-regular\ graphs$. An inverse transformation that faithfully recovers the time series is not possible for visibility graphs. Since visibility graphs is a qualitative methodology, the values of a time series derived from these graphs would vary in an interval, whose length would be different for each point in the time series. The framework of complex networks for analyzing heart rate variability data towards the detection of early warnings and the design of clinical tools for disease management has been considered before as other nonlinear methods [19]. Visibility graphs have been applied to

the analysis of congestive heart failure [20]. Inhere, a statistical analysis of the scale-freeness of the obtained network is used for the detection of early stages of the disease. In a broader analysis, several summary statistics of a horizonal visibility network have been proposed as useful for the analysis of heart rate variability [21]. In general, the use of summary statistics for the detection of early warnings in a transition of dynamical state may be difficult since such statistics may rely on inadequate data or other factors [22]. Other studies have shown that the incorporation of respiration signals to the electrocardiogram data increase the detection of a VT episode [23]. Hitherto, the effect of the vagus nerve in the heart activity has been recently investigated [25]. Different techniques based on other methodologies and data have shown different times before the VT episode occurs [23, 24]. Any predictor, regardless of the methodology must clearly distinguish a VT episode from the usual cardiac arrythmias of each patient to avoid false positive detection. So far, the low heart rate variability has been considered as the single predictor of heart failure, although the forces for the acceleration and deceleration in heart activity have been shown to be uncoupled [25]. The device used by the patient has high impact on any method for the detection of early warnings of a cardiac malfunction, as it has been shown that ICDs can detect QT variability in near-field or far-field right ventricular intracardiac electrogram [26]. ICD are excellent machines devoted to terminating VT and they have proved its efficacy to prevent sudden cardiac death in different clinical settings. The performance of the algorithms has been tested first to detect the VT and to provide appropriate shocks. Then, algorithms were improved to avoid unnecessary ("inappropriate") discharges to the patient. In recent years there has been a small but strong movement in the medical community towards the possibility of alerting the patient when an ICD shock is going to occur. This possibility is not minor. From a clinical point of view, such alert could permit the patient or his close relatives to take appropriate measures before the shock takes place. A new window of opportunity (clinical interventions) could be generated if a software could be able to detect with some seconds or, even better, minutes, the possibility of an imminent ICD shock. Until today, there is no such possibility. The present retrospective study sheds light of a possible mathematical analysis that could detect "early warnings" of an appropriate ICD shock for VT with an average of 8:30 minutes. The process of optimization of the $\varepsilon$ parameter value requires a more extensive clinical experimentation. It stands to reason that the parameter value is specific for each individual patient and it ought to be tuned from the complete set of clinical parameters of the patient to avoid false-positives and false-negatives. In a more general case, it is also probable that the parameter value of $\varepsilon$ is not fixed throughout the day and is dependent of the circadian rhythm of the patient.

Obviously, this mathematical application should be tested prospectively, but this can only be done if implemented into the software of the ICD. Collaboration with ICD industry is vital to achieve such goal.

## Conclusions

Early warnings of the VA episode could be detected using their corresponding $\varepsilon-regular$ *graphs*, even 8:30 minutes before the ICD comes into action. A prospective study is warranted to further corroborate this finding.

## Supporting information

**S1 File. The script was developed using Wolfram Mathematica 12.0.**
(NB)

**S1 Fig. The time series of the R-R intervals for the VT episode are substracted from the control group from an average of the records of 81 patients.** The regular graphs are derived considering $\varepsilon$ values of 0.01, 0.02, 0.03, 0.04, 0.05, 0.06 and a window of 60 points. The degree centrality is averaged for each window and plotted in this figure.
(TIFF)

## Acknowledgments

We thank Juan R. Bobadilla for technical computer support.

## Author Contributions

**Conceptualization:** Gabriel S. Zamudio, Marco V. José.

**Data curation:** Gabriel S. Zamudio.

**Formal analysis:** Gabriel S. Zamudio, Marco V. José.

**Funding acquisition:** Marco V. José.

**Investigation:** Gabriel S. Zamudio, Manlio F. Márquez, Marco V. José.

**Methodology:** Gabriel S. Zamudio, Marco V. José.

**Project administration:** Marco V. José.

**Resources:** Marco V. José.

**Software:** Gabriel S. Zamudio.

**Supervision:** Manlio F. Márquez, Marco V. José.

**Validation:** Gabriel S. Zamudio, Manlio F. Márquez.

**Visualization:** Manlio F. Márquez.

**Writing – original draft:** Gabriel S. Zamudio, Marco V. José.

**Writing – review & editing:** Gabriel S. Zamudio, Manlio F. Márquez, Marco V. José.

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
