## [Decision Letter · Decision Letter 0]

29 Jul 2020

PONE-D-20-17194

Anticipation of ventricular tachyarrhythmias by a novel mathematical method: Further insights towards an early warning system in implantable cardioverter defibrillators

PLOS ONE

Dear Dr. José,

Thank you for submitting your manuscript to PLOS ONE. After careful consideration, we feel that it has merit but does not fully meet PLOS ONE’s publication criteria as it currently stands. Therefore, we invite you to submit a revised version of the manuscript that addresses the points raised during the review process.

Please address comments indicated by the Reviewers.

We look forward to receiving your revised manuscript.

Kind regards,

Elena G. Tolkacheva, PhD

Academic Editor

PLOS ONE

Journal Requirements:

2. Please ensure that you have reported at the beginning of your methods or results section the key performance measures that were used to establish validity and utility of your algorithm.

Reviewers' comments:

Reviewer's Responses to Questions

**Comments to the Author**

1. Is the manuscript technically sound, and do the data support the conclusions?

Reviewer #1: Yes

Reviewer #2: Yes

2. Has the statistical analysis been performed appropriately and rigorously? 

Reviewer #1: Yes

Reviewer #2: Yes

3. Have the authors made all data underlying the findings in their manuscript fully available?

Reviewer #1: Yes

Reviewer #2: Yes

4. Is the manuscript presented in an intelligible fashion and written in standard English?

Reviewer #1: Yes

Reviewer #2: Yes

5. Review Comments to the Author

Reviewer #1: This is an interesting mathematical approach to look at heart rate variability and its ability to predict VT events. The analytical approach is sufficiently novel to be of interest, and it appears to show an ability to discriminate RR interval time series preceding VT events from control RR interval time series from the same patients. However, we are only really shown a representative example of time series analysis and associated derived metrics (Figure 3) and there is no tabulation or graphical representation of this for all 81 patients. So it is not clear that this is genuinely a "representative" result, or the single best time series in which discrimination occurred. Presentation of the results from the whole cohort would address this.

The discussion should note that a significant clinical limitation of the work is that this type of approach is unlikely to work in patients with AF. Because the AF group with ICDs is currently a significant challenge to us clinically, this is an important limitation.

Reviewer #2: In this manuscript, the authors developed a method to transform time series into graphs which is sensitive to a single parameter. And they also demonstrated the feasibility of using this method to detect early warning of VA episode. The followings are my comments to further improve this manuscript.

1. In figure 1, the authors gave an example to using �-regular graph and compare their method with visibility graph. The author could also show the result of visibility graph in figure 1 to make the comparison more straightforward.

2. From figure 2, the authors concluded “The degree distributions for the graphs constructed form the Brownian motion approximates a Gaussian curve despite its multiple modes; and the distribution from points sampled from a normal distribution approximates a lognormal distribution.” The authors simply said what they observed in the plot. They should further explain the results.

3. The authors show the complete data in supplementary files instead of providing the website source.

4. In figure 3, the authors only showed the graph with ����������why is this? They should sweep this parameter and show how the graph looks like with different parameter values. They should also explain how they optimize this parameter.

5. “We propose this change in the dynamic of the R-R intervals as a measurable and detectable early warning of the VT event, occurring an average of 514.625 seconds (8:30 minutes) before the start of the VT episode.” How did the authors get this number (514.625) from the graph? They should provide the details.

6. PLOS authors have the option to publish the peer review history of their article (what does this mean?). If published, this will include your full peer review and any attached files.

Reviewer #1: No

Reviewer #2: No

---

## [Author Response · Author response to Decision Letter 0]

13 Aug 2020

August 10th, 2020

Elena G. Tolkacheva, PhD

Academic Editor

PLOS ONE

Journal: PLOS ONE

Manuscript ID: PONE-D-20-17194

Title: Anticipation of ventricular tachyarrhythmias by a novel mathematical method: Further insights towards an early warning system in implantable cardioverter defibrillators

Authors: Gabriel S. Zamudio, Manlio F. Márquez, Marco V. José

Dear Dr. Tolkacheva:

Thank you for your email dated July 29th, 2020 and for the reviewer’s comments. We are thankful to the reviewers for their helpful criticisms. We carefully evaluated and answered all the questions posed by reviewers. Please check below.

Answer to reviewers

Reviewers' comments:

Reviewer's Responses to Questions

Comments to the Author

1. Is the manuscript technically sound, and do the data support the conclusions?

Reviewer #1: Yes

Reviewer #2: Yes

2. Has the statistical analysis been performed appropriately and rigorously?

Reviewer #1: Yes

Reviewer #2: Yes

3. Have the authors made all data underlying the findings in their manuscript fully available?

Reviewer #1: Yes

Reviewer #2: Yes

4. Is the manuscript presented in an intelligible fashion and written in standard English?

Reviewer #1: Yes

Reviewer #2: Yes

5. Review Comments to the Author

Reviewer #1: This is an interesting mathematical approach to look at heart rate variability and its ability to predict VT events. The analytical approach is sufficiently novel to be of interest, and it appears to show an ability to discriminate RR interval time series preceding VT events from control RR interval time series from the same patients. However, we are only really shown a representative example of time series analysis and associated derived metrics (Figure 3) and there is no tabulation or graphical representation of this for all 81 patients. So it is not clear that this is genuinely a "representative" result, or the single best time series in which discrimination occurred. Presentation of the results from the whole cohort would address this.

The result of the analysis of the 81 patients can be seen in this plot:

There is a great variability among each individual patient.

In Discussion we added the following paragraph:

“The process of optimization of the epsilon parameter value requires a more extensive clinical experimentation. It stands to reason that the parameter value is specific for each individual person and ought to be tuned from the complete set of clinical parameters of the patient to avoid false-positives and false-negatives. In a more general case, it is also probable that the parameter value is not fixed throughout the day and is dependent of the circadian rhythm of the patient.”

The discussion should note that a significant clinical limitation of the work is that this type of approach is unlikely to work in patients with AF. Because the AF group with ICDs is currently a significant challenge to us clinically, this is an important limitation.

Thank you very much for pointing out this important limitation. We have now included a comment in Limitations as follows: 

“A significant clinical limitation of this work is the fact that this approach is restricted to patients with normal sinus rhythm and is unlikely to work in patients with atrial fibrillation or those with a pacemaker. In the former group because of the large variability of R-R intervals, and in the later because of fixed pacing rhythms.”

Reviewer #2: In this manuscript, the authors developed a method to transform time series into graphs which is sensitive to a single parameter. And they also demonstrated the feasibility of using this method to detect early warning of VA episode. The followings are my comments to further improve this manuscript.

1. In figure 1, the authors gave an example to using �-regular graph and compare their method with visibility graph. The author could also show the result of visibility graph in figure 1 to make the comparison more straightforward.

We modified Figure 1 to include Figure 1C that shows the transformation of the time series using the visibility graph algorithm.

2. From figure 2, the authors concluded “The degree distributions for the graphs constructed form the Brownian motion approximates a Gaussian curve despite its multiple modes; and the distribution from points sampled from a normal distribution approximates a lognormal distribution.” The authors simply said what they observed in the plot. They should further explain the results.

We have now included a comment in Methods as follows: 

“The distributions of the degree measurements maintain, up to some extent, the statistical properties from the time series they were derived from. The difference between any two values of the Brownian motion follows a normal distribution, and this property is shown by the epsilon-regular graphs in the degree centrality measure when setting the epsilon parameter equal to values related to the standard deviation.”

3. The authors show the complete data in supplementary files instead of providing the website source.

We now provide the website source of the data in the format requested by the journal guidelines, also, the URL of the websites are cited in the section Application to Imminent VT.

4. In figure 3, the authors only showed the graph with ����������why is this? They should sweep this parameter and show how the graph looks like with different parameter values. They should also explain how they optimize this parameter.

The parameter epsilon������� showed the best results for the current application. We have added the experimental results for the parameter values of �psilon������1, 0.02, 0.03, 0.05, and 0.06 in Supporting Information II. We have added the following paragraph in Application to Imminent VT section:

“The optimization of the parameter �psilon�is based on the statistical parameters of the R-R time series. The average, minimum and maximum distance between two points of the R-R time series are: 0.043, 1.11x10^(-16), 0.12, respectively. Thus, the �psilon value of 0.04 approximates the mean of the differences. In Supporting Information II, we show the degree time series when the values ����1, 0.02, 0.03, 0.04, 0.05, and 0.06 are assigned to the parameter epsilon (Fig S1). Note that the overall pattern is preserved. In particular, the abrupt change of the dynamics is captured by the different values of the parameter epsilon”.

5. “We propose this change in the dynamic of the R-R intervals as a measurable and detectable early warning of the VT event, occurring an average of 514.625 seconds (8:30 minutes) before the start of the VT episode.” How did the authors get this number (514.625) from the graph? They should provide the details.

In Application to Imminent VT we added the following sentence:

“The count of 514.625 seconds corresponds to the sum of the average of the time lapses of the last R-R intervals starting from the point 276 to when the VT episode begins at point 985.”

6. PLOS authors have the option to publish the peer review history of their article (what does this mean?). If published, this will include your full peer review and any attached files.

Do you want your identity to be public for this peer review? For information about this choice, including consent withdrawal, please see our Privacy Policy.

Reviewer #1: No

Reviewer #2: No

All in all, we do thank the reviewers for their helpful comments and criticisms; for we feel that they have helped us to improve the quality and presentation of the paper. We expect that the present version of the manuscript answers all their concerns.

Yours Sincerely,

Marco V. José PhD

Theoretical Biology Group

Instituto de Investigaciones Biomédicas

Universidad Nacional Autónoma de México

Apartado Postal 70228 Ciudad Universitaria

C.P. 04510 CDMX, México

Tel: 52-55-5622-3894

Mobile: 52-55-1320-6111

Author for correspondence: Marco V. José (marcojose@biomedicas.unam.mx)

---

## [Decision Letter · Decision Letter 1]

15 Sep 2020

Anticipation of ventricular tachyarrhythmias by a novel mathematical method: Further insights towards an early warning system in implantable cardioverter defibrillators

PONE-D-20-17194R1

Dear Dr. José,

We’re pleased to inform you that your manuscript has been judged scientifically suitable for publication and will be formally accepted for publication once it meets all outstanding technical requirements.

Kind regards,

Elena G. Tolkacheva, PhD

Academic Editor

PLOS ONE

Additional Editor Comments (optional):

Reviewers' comments:

Reviewer's Responses to Questions

**Comments to the Author**

1. If the authors have adequately addressed your comments raised in a previous round of review and you feel that this manuscript is now acceptable for publication, you may indicate that here to bypass the “Comments to the Author” section, enter your conflict of interest statement in the “Confidential to Editor” section, and submit your "Accept" recommendation.

Reviewer #1: All comments have been addressed

Reviewer #2: All comments have been addressed

2. Is the manuscript technically sound, and do the data support the conclusions?

Reviewer #1: Yes

Reviewer #2: (No Response)

3. Has the statistical analysis been performed appropriately and rigorously? 

Reviewer #1: Yes

Reviewer #2: (No Response)

4. Have the authors made all data underlying the findings in their manuscript fully available?

Reviewer #1: Yes

Reviewer #2: (No Response)

5. Is the manuscript presented in an intelligible fashion and written in standard English?

Reviewer #1: Yes

Reviewer #2: (No Response)

6. Review Comments to the Author

Reviewer #1: Thanks for your responses to my previous comments on your manuscript. I really liked the figure of all 81 patients data - I thought that looked amazing, but I agree it isn't really suitable for inclusion within the manuscript. The comments within the discussion address my concerns well.

Reviewer #2: (No Response)

7. PLOS authors have the option to publish the peer review history of their article (what does this mean?). If published, this will include your full peer review and any attached files.

Reviewer #1: No

Reviewer #2: No

---

## [Editor Report · Acceptance letter]

17 Sep 2020

PONE-D-20-17194R1 

Anticipation of ventricular tachyarrhythmias by a novel mathematical method: Further insights towards an early warning system in implantable cardioverter defibrillators 

Dear Dr. José:

I'm pleased to inform you that your manuscript has been deemed suitable for publication in PLOS ONE. Congratulations! Your manuscript is now with our production department. 

Kind regards, 

on behalf of

Dr. Elena G. Tolkacheva 

Academic Editor

PLOS ONE